# Autophagy Modulation in Human Thyroid Cancer Cells following Aloperine Treatment

**DOI:** 10.3390/ijms20215315

**Published:** 2019-10-25

**Authors:** Hui-I Yu, Hui-Ching Shen, Shu-Hsin Chen, Yun-Ping Lim, Hsiang-Hsun Chuang, Tsai-Sung Tai, Fang-Ping Kung, Chieh-Hsiang Lu, Chia-Yi Hou, Ying-Ray Lee

**Affiliations:** 1Division of Endocrinology and Metabolism, Department of Internal Medicine, Ditmanson Medical Foundation Chiayi Christian Hospital, Chiayi City 600, Taiwan; 04490@cych.org.tw (H.-I.Y.); 04486@cych.org.tw (H.-H.C.); 04015@cych.org.tw (T.-S.T.); 07266@cych.org.tw (F.-P.K.); 02602@cych.org.tw (C.-H.L.); 2Department of Clinical Pathology, Chi Mei Medical Center, Liouying 736 Taiwan; huiching0105@gmail.com (H.-C.S.); 960360@mail.chimei.org.tw (C.-Y.H.); 3Department of Medical Research, Ditmanson Medical Foundation Chiayi Christian Hospital, Chiayi City 600, Taiwan; 10472@cych.org.tw; 4Department of Pharmacy, College of Pharmacy, China Medical University, Taichung 404, Taiwan; limyp@mail.cmu.edu.tw

**Keywords:** aloperine, thyroid cancer, autophagy

## Abstract

Aloperine, an alkaloid isolated from *Sophora*
*alopecuroides*, exhibits multiple pharmacological activities including anti-inflammatory, antioxidant, antiallergic, antinociceptive, antipathogenic, and antitumor effects. Furthermore, it exerts protective effects against renal and neuronal injuries. Several studies have reported antitumor effects of aloperine against various human cancers, including multiple myeloma; colon, breast, and prostate cancers; and osteosarcoma. Cell cycle arrest, apoptosis induction, and tumorigenesis suppression have been demonstrated following aloperine treatment. In a previous study, we demonstrated antitumor effects of aloperine on human thyroid cancer cells through anti-tumorigenesis and caspase-dependent apoptosis induction via the Akt signaling pathway. In the present study, we demonstrated the modulation of the autophagy mechanism following the incubation of multidrug-resistant papillary and anaplastic human thyroid cancer cells with aloperine; we also illustrate the underlying mechanisms, including AMPK, Erk, JNK, p38, and Akt signaling pathways. Further investigation revealed the involvement of the Akt signaling pathway in aloperine-modulated autophagy in human thyroid cancer cells. These results indicate a previously unappreciated function of aloperine in autophagy modulation in human thyroid cancer cells.

## 1. Introduction

Aloperine, a quinolizidine alkaloid isolated from *Sophora alopecuroides*, reportedly exhibits multiple pharmacological activities, including anti-inflammatory, antiallergenic, antiviral, antimicrobial, antinociceptive, and antitumor effects; moreover, it exhibits protective effects against renal and neuronal injuries as well as pulmonary fibrosis [1,2,3,4,5,6,7,8,9,10,11,12,13,14,15,16,17,18,19,20]. Aloperine exerts antitumor effects against multiple human cancers, including multiple myeloma; colon, breast, prostate, and thyroid cancers; and osteosarcoma [1,7,12,15,18,20]. In previous studies, our and other research groups have demonstrated that aloperine can suppress tumorigenesis, inhibit tumor cell proliferation, and induce cell cycle arrest and apoptosis in various human cancer cells [1,7,12,15,18,19]. Studies investigating the mechanisms underlying the antitumor activity of aloperine have demonstrated the modulation of various signaling pathways in human tumors, including JAK/STAT3, Ras-Erk, and PI3K/Akt pathways, following aloperine treatment [1,7,13,15,19,20]. Furthermore, an in vivo study has demonstrated the safety and efficacy of aloperine as a therapeutic agent [7]. Although antitumor activities of aloperine have been reported in various studies, its detailed mechanism of action and possible further anticancer effects should be elucidated urgently.

Autophagy, a self-degradative mechanism, disassembles unnecessary or dysfunctional components in cells, thereby maintaining homeostasis and intracellular energy balance. Under stressful conditions, such as nutrient deprivation, hypoxia, or infection, autophagy is induced to eliminate stress and maintain homeostasis for cell survival [21,22]. During autophagy activation, double-membrane vesicles (autophagosome or autophagic vacuole) are formed and specific or non-specific target cargos are recruited in the cytoplasm for disassembling and recycling unnecessary or dysfunctional cellular components via lysosomes (autolysosomes) [22]. Currently, autophagy is implicated in various physiological responses and pathophysiological processes, including aging, metabolic disorders, neurodegenerative diseases, cardiovascular disorders, immune responses, carcinogenesis, and infectious diseases [23]. Autophagy plays a dual role in cancer. It can promote cancer growth and survival by maintaining cellular energy production and eliminating stress, but it has also been recognized as a therapeutic strategy against cancer [24]. However, there is insufficient evidence regarding the modulation of autophagy by aloperine. Therefore, the regulation of autophagy machinery and related mechanisms following aloperine treatment are worth exploring.

We have previously demonstrated anticancer activity of aloperine in human thyroid cancer cells through elevated caspase-dependent apoptosis via the PI3K/Akt signaling pathway [15]. The PI3K/Akt/mTOR signaling pathway plays important roles in autophagy regulation [25] and is a potential target for anticancer therapy. In the present study, we evaluated autophagy in multiple human thyroid cancer cells following aloperine treatment. Furthermore, we investigated the upstream signaling pathway involved in aloperine-mediated autophagy modulation.

## 2. Results

### 2.1. Aloperine Reduces Cellular Viability in Human Thyroid Cancer Cells

Various studies have demonstrated that aloperine can induce apoptosis in human cancer cells, including those of multiple myeloma; osteosarcoma; and hepatocellular, breast, colon, thyroid, and prostate cancer [1,7,12,15,18,19]. In the present study, we confirmed the effects of aloperine on cellular viability in undifferentiated human thyroid cancer cells as well as in multidrug-resistant anaplastic and papillary thyroid cancer cells. The viability of KMH-2, 8505c, and IHH-4 cells treated with various concentrations of aloperine was significantly reduced in a dosage-dependent manner compared with that of the control cells (Figure 1). At 48 h post-incubation with aloperine, 50% cytotoxic concentration (CC_50_) for KMH-2, 8505c, and IHH-4 cells was 207.3, 268.4, and 169.4 µM, respectively, suggesting that KMH-2 and IHH-4 cells are more sensitive to aloperine comparing with 8505c cells. Whether this difference in these cells are due to genetic background or other reasons warrants further investigation.

### 2.2. Aloperine Promotes Autophagy Activation in Human Thyroid Cancer Cells

To evaluate whether aloperine can modulate cellular autophagy activity, human thyroid cancer cells were treated with aloperine at various concentrations. The expression of LC3-II, a biomarker for autophagosome formation, was determined using western blot. Elevated LC3-II expression was observed in KMH-2, 8505c, and IHH-4 cells following aloperine treatment (Figure 2). Moreover, LC3-II and p62 expression was decreased in KMH-2 and IHH-4 cells following treatment with aloperine at a high concentration or for a long time (Figure 2A,B,E,F), suggesting that aloperine induced autophagic flux in these cells. However, LC3-II and p62/SQSTM1 (p62) expression continued to increase in 8505c cells (Figure 2C,D), suggesting that aloperine blocked autophagic flux in these cells. Furthermore, immunofluorescence staining confirmed aloperine-induced autophagosome formation in KMH-2 and IHH-4 cells. Rapamycin elevated LC3 puncta in KMH-2 and IHH-4 cells (Figure 3). Aloperine treatment induced LC3 puncta, and co-incubation with 3-MA could reduce aloperine-mediated LC3 puncta (Figure 3) in KMH-2 and IHH-4 cells, suggesting that aloperine treatment promoted autophagosome formation in these cells following. To confirm the induction of autophagic flux in KMH-2 and IHH-4 cells by aloperine, the mRFP-EGFP-LC3 vector was transfected into KMH-2 cells and autophagosome and autolysosome puncta were located using confocal microscopy. Aloperine treatment promoted autophagosome formation and simultaneously enhanced autophagic flux in KMH-2 and IHH-4 cells (Figure 4). Moreover, chloroquine treatment could block the aloperine-mediated induction of autophagic flux. These results demonstrate that aloperine acts as an autophagy inducer in KMH-2 and IHH-4 cells but blocks autophagic flux in 8505c cells.

### 2.3. Modulations of Signaling Pathways with Aloperine in Human Thyroid Cancer Cells

To investigate the mechanisms underlying aloperine-mediated autophagy induction, the expression and activation of AMPK, Akt/mTOR, Erk, p38, and JNK signaling pathways were determined in KMH-2 and IHH-4 cells following aloperine treatment. The expression and activation of the AMPK pathway was reduced in both KMH-2 and IHH-4 cells (Appendix A). Moreover, the activation of the Akt/mTOR and p70S6K pathways decreased in an aloperine dosage-related manner treatment in both cell lines, whereas the expression of LC3-II increased (Figure 5A). In addition, aloperine treatment suppressed overall mTOR expression (Figure 5A), suggesting that aloperine regulates mTOR translation. In addition, the expression of phospho-p38 and phospho-Erk was reduced (Figure 5B), whereas that of LC3-II was increased; however, the expression of phospho-JNK remained unchanged. These results suggest that aloperine modulates the Akt/mTOR, Erk, and p38 signaling pathways and activates autophagy in KMH-2 and IHH-4 cells.

### 2.4. Akt Signaling Pathway Contributes to Aloperine-mediated Autophagy Induction in Human Thyroid Cancer Cells

To address whether the Akt/mTOR, p38, and Erk signaling pathways are involved in the aloperine-mediated autophagy, KMH-2 and IHH-4 cells were pre-incubated with inhibitors including perifosine, SB203580, and PD98059. Inhibiting Akt pathway activation increased LC3-II expression in KMH-2 and IHH-4 cells (Figure 5A); this was also observed in aloperine-treated groups. Moreover, compared with aloperine alone, the combination of aloperine and perifosine suppressed phospho-Akt and increased LC3-II expression to a greater extent (Figure 6A), suggesting that Akt pathway inhibition contributes to aloperine-mediated autophagy induction. Moreover, treatment with SB203580 and PD98059 inhibited p38 and Erk pathway activation in KMH-2 and IHH-4 cells (Figure 6B,C), and aloperine treatment suppressed p38 and Erk pathway activation. Furthermore, combination of aloperine with PD98059 or SB203580 decreased LC3-II expression in KMH-2 and IHH-4 cells (Figure 6B,C), suggesting that aloperine-mediated autophagy activation is not regulated via Erk and p38 pathway modulation. However, the physiological significance of aloperine-mediated decrease in phospho-Erk and phospho-p38 warrants further investigation. Meanwhile, we confirmed autophagosome puncta formation in cells treated with a combination of aloperine and perifosine. Perifosine-increased Akt pathway inhibition significantly increased LC3 puncta formation (Figure 7). In addition, we confirmed the role of the Akt pathway in aloperine-mediated autophagy induction; a constitutively active-form construction of Akt was transiently transfected into KMH-2 and IHH-4 cells, and the activation of Akt and expression of LC3-II were monitored using Western blotting. The overexpression of active-form Akt in the cells treated with aloperine reduced the expression of LC3-II, demonstrating that the Akt signaling pathway is the upstream pathway involved in aloperine-mediated autophagy induction (Figure 8). These results demonstrate that aloperine may serve as an autophagy inducer via Akt/mTOR signaling pathway suppression in human thyroid cancer cells.

### 2.5. Aloperine-Mediated Autophagy Exerts a Cytotoxic Effect in Human Thyroid Cancer Cells

To determine the role of aloperine-mediated autophagy modulation in human thyroid cancer cells, KMH-2 and IHH-4 cells were incubated with aloperine or combination with 3-MA or rapamycin, and the cellular viability was examined with CCK-8 assay. The cell viability was significantly reduced in the cells under aloperine treatment (Figure 9). However, the cellular toxicity was rescued in the cells under aloperine combination with 3-MA treatment (Figure 9). On the other hand, cells incubated with aloperine and rapamycin showed higher cytotoxicity (Figure 9). These data suggested that aloperine-mediated autophagy serve a cytotoxic activity in KMH-2 and IHH-4 cells.

## 3. Discussion

Aloperine, a natural alkaloid isolated from the herb *S. alopecuroides*, has been reported to exhibit anticancer activity in various human cancers [1,7,12,15,18,20]. Previous studies have demonstrated that aloperine mediates cell proliferation, cell cycle inhibition and regulation, apoptosis induction, and tumor migration/invasion and suppression [1,7,12,15,18,19,26]. In a previous study, we demonstrated that anticancer effects of aloperine in multiple human thyroid cancer cells via the suppression of cell proliferation and tumorigenesis as well as induction of cell cycle arrest and apoptosis [15]. In this study, we evaluated autophagy regulation in IHH-4 (a multidrug-resistant papillary thyroid carcinoma), 8505C (an undifferentiated thyroid carcinoma), and KMH-2 (a multidrug-resistant anaplastic thyroid carcinoma) cells treated with aloperine. We also explored the mechanisms involved in aloperine-modulated autophagy activation.

We demonstrated that aloperine activates the autophagy machinery, promotes autophagosome formation, and increases autophagic flux in KMH-2 and IHH-4 cells (Figure 2A,B,E,F, Figure 3, Figure 4), suggesting its role as an autophagy inducer in these cells. Although LC3-II and p62 expression levels were elevated in 8505c cells following aloperine treatment, these levels did not further decrease with prolonged treatment (Figure 2C,D), suggesting that aloperine acts as an autophagy inhibitor in 8505c cells. However, further investigation is required to determine whether aloperine can simultaneously activate autophagy and block autophagic flux in 8505c cells. Alternatively, whether aloperine alters lysosomal functions, inhibits lysosomal proteolysis, or blocks the delivery of cargo to lysosomes needs further evaluation.

The PI3K/Akt/mTOR, AMPK, and MAPK/Erk signaling pathways play major roles in autophagy induction [27]. In this study, we evaluated the expression and activation of AMPK, Akt/mTOR, Erk, p38, and JNK pathways in cells following aloperine treatment. We demonstrated that the aloperine-suppressed Akt/mTOR signaling pathway is the upstream mechanism for autophagy induction in KMH-2 and IHH-4 cells (Figure 5, Figure 6, Figure 7 and Figure 8). Although our results show that aloperine can suppress the activation of the p38 and Erk pathways in KMH-2 and IHH-4 cells (Figure 5A,B), this suppression with a combination of aloperine and SB203580 or PD98059 treatment did not significantly increase LC3-II expression (Figure 6B,C). These results suggest that the aloperine-mediated inhibition of p38 and Erk pathways is not the underlying mechanism for aloperine-mediated autophagy induction. However, the physiological significance of aloperine-inhibited p38 and Erk signaling pathways in KMH-2 and IHH-4 cells warrants further investigation. In addition, we cannot rule out whether factors other than the Akt pathway are involved in aloperine-mediated autophagy induction.

Autophagy is a self-degradation process that clears unnecessary intracellular organelles and proteins, playing an important role in cellular homeostasis [22,23,27]. Multiple compounds including clinical chemotherapeutic agents and/or natural products induce autophagy. However, such autophagy induction is either cytotoxic or cytoprotective [28], demonstrating that autophagy induction in cancer cells may be more complicated. Although excessive and sustained autophagy may lead to cell death and tumor shrinkage, and autophagic cell death has been reported in numerous reports, the cytotoxic role of autophagy remains debatable because of insufficient data on autophagic cell death markers [29]. In the present study, we demonstrated that aloperine can modulate the autophagy machinery and induce autophagosome as well as autophagic flux in human thyroid cancer cells. Blocking of autophagy with 3-MA or enhancing of autophagy with rapamycin in the aloperine-treated cells could reduce or enhance aloperine-mediated cytotoxicity. Therefore, we suggested that aloperine-mediated autophagy exerts a cytotoxic role in human thyroid cancer cells.

Autophagy activation by natural products has been considered a double-edged sword in determining the cell fate of human cancers, and the interplay between autophagy and apoptosis has recently been highlighted [30]. Several proteins and signaling pathways, including the p53, Bcl-2, DAPK, Akt/mTOR, and JNK pathways, act as scaffolds in mediating the crosstalk between autophagy and apoptosis [30,31]. In a previous study, we reported that aloperine induces caspase-dependent apoptosis through PI3K/Akt inhibition pathway in human thyroid cancer cells [15]. In the present study, cells treated with aloperine exhibited autophagy induction through Akt/mTOR pathway suppression in human thyroid cancer cells. Therefore, we speculate that Akt/mTOR pathway inhibition induces both apoptosis and autophagy in human thyroid cancer cells following aloperine treatment. Overall, our findings indicate interplay of molecules in human thyroid cancer cells following aloperine treatment, which mediates both autophagy and apoptosis, and these molecules may be effective targets in developing anticancer therapies. In addition, aloperine exhibits multiple pharmacological activities, including anti-inflammatory, antiallergenic, antiviral, antimicrobial, and antinociceptive effects against renal and neuronal injuries as well as pulmonary fibrosis [1,2,3,4,5,6,7,8,9,10,11,12,13,14,15,16,17,18,19,20]. Here, we demonstrate autophagy regulation by aloperine. Whether aloperine can protect against renal and neuronal injuries as well as pulmonary fibrosis through autophagy induction should be an interesting topic to explore in the future [32,33].

## 4. Materials and Methods

### 4.1. Cell Line and Cell Culture

Human thyroid cancer cell lines including KMH-2 (multidrug-resistant anaplastic thyroid carcinoma), 8505c (undifferentiated thyroid carcinoma), and IHH-4 (multidrug-resistant papillary thyroid carcinoma) were purchased from Japan Collection of Research Bioresources Cell Bank (JCRB, Japan). IHH-4 and KMH-2 cells were cultured in DMEM + RPMI (1:1) medium (GIBCO, Gaithersburg, MD, USA), and 8505c cells were cultured in MEM medium (GIBCO) supplemented with 10% FBS (Biological Industries, Kibbutz Beit Haemek, Israel) at 37 °C in a 5% CO_2_ incubator.

### 4.2. Cell Viability Assay

Cells (5 × 10^3^ cells/well) were seeded in 96-well cell culture plates, and adherent cells were incubated in a control medium containing 0.01% dimethyl sulfoxide (DMSO) or a medium containing aloperine (Selleck Chemicals, Houston, TX, USA). Cell viability was examined using the CCK-8 assay kit (Enzo Life Sciences, Farmingdale, NY, USA) following treatment for indicated time and at indicated dosages. Three independent experiments were performed.

### 4.3. Autophagosome Detection

Cells (without starvation) were treated with or without aloperine for various durations. The expression of autophagy marker LC3-II was examined using Western blotting. Furthermore, autophagosome formation was detected using immunofluorescence staining under a laser confocal scanning microscope (LSM800, ZEISS, Germany). To determine autophagic flux induction following aloperine treatment, p62 expression in cells was examined. GAPDH was used as a loading control in western blotting, and DAPI was used to label the nucleus.

### 4.4. Western Blotting

Cells were cultured in 10-cm cell culture dishes and treated with aloperine. DMSO was used as a negative control. The whole cellular extract was subjected to sodium dodecyl sulfate–polyacrylamide gel electrophoresis, and the separated proteins were electrically transferred to a PVDF membrane (Millipore Corporation, USA). The membrane was blocked with primary antibodies (anti-LC3 Ab; Abcam, USA; anti-GAPDH Ab; GeneTex, USA; anti-AMPK α-1 Ab; Cell signaling, USA; anti-phosphorylated-AMPK α-1 (Thr 172) Ab; Cell Signaling, USA; anti-Akt Ab; Santa Cruz, USA; anti-phosphorylated-Akt (Ser 473) Ab; Santa Cruz, USA; anti-mTOR Ab; Cell Signaling, USA; anti-phosphorylated-mTOR (Ser 2448) Ab; Cell Signaling, USA; anti-p70S6K Ab; Cell Signaling, USA; anti-phosphorylated-p70S6K (Thr 389) Ab; Cell Signaling; anti-p62/SQSTM1 Ab; Abgent, USA; anti-Erk Ab; Cell Signaling, USA; anti-phosphorylated-Erk (Thr 202/204) Ab; Cell Signaling, USA; anti-JNK Ab; Cell Signaling, USA; anti-phosphorylated-JNK (Thr 183/Tyr 185) Ab; Cell Signaling, USA; anti-p38 Ab; Cell Signaling, USA; and anti-phosphorylated (Thr 180/Tyr 182) Ab; Cell Signaling Ab, USA) and was analyzed using the BioSpectrum 800 Imaging System (UVP, CA, USA).

### 4.5. Plasmid Transfection

To observe autophagosome and autolysosome formation in cells treated with aloperine, a pmRFP-EGFP-LC3 plasmid (purchased from Addgene, Watertown, MA, USA) [34] was transfected with Lipofectamine 2000 (Thermo Scientific, Waltham, MA, USA) according to the manufacturer’s instructions. To confirm the role of the Akt pathway in aloperine-mediated autophagosome induction, the pHRIG-Akt1 plasmid (a constitutively active-form construct of human Akt1, a myristoylated form of Akt-1, purchased from Addgene) and pBSSK^+^ (an empty vector used as a negative control) were transfected with Lipofectamine 2000 (Thermo Scientific, Waltham, MA, USA) according to the manufacturer’s instructions.

### 4.6. Statistical Analysis

Data are presented as mean and standard error. Statistical differences were analyzed using one-way analysis of variance and Fisher’s least significant difference test. Statistical significance was defined as *p* < 0.05.

## Figures and Tables

**Figure 1 ijms-20-05315-f001:**
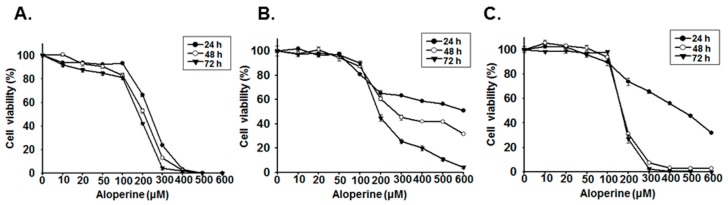
Suppression of human thyroid cancer cell growth following aloperine treatment. (**A**) KMH-2, (**B**) 8505c, and (**C**) IHH-4 cells were treated with aloperine at various dosages, and their viabilities were determined after 24, 48, and 72 h post-treatment using the CCK-8 assay. Cells treated with dimethyl sulfoxide (DMSO) were used as a negative control, and all groups were normalized with the control group. The results are expressed as mean ± SD of three independent experiments.

**Figure 2 ijms-20-05315-f002:**
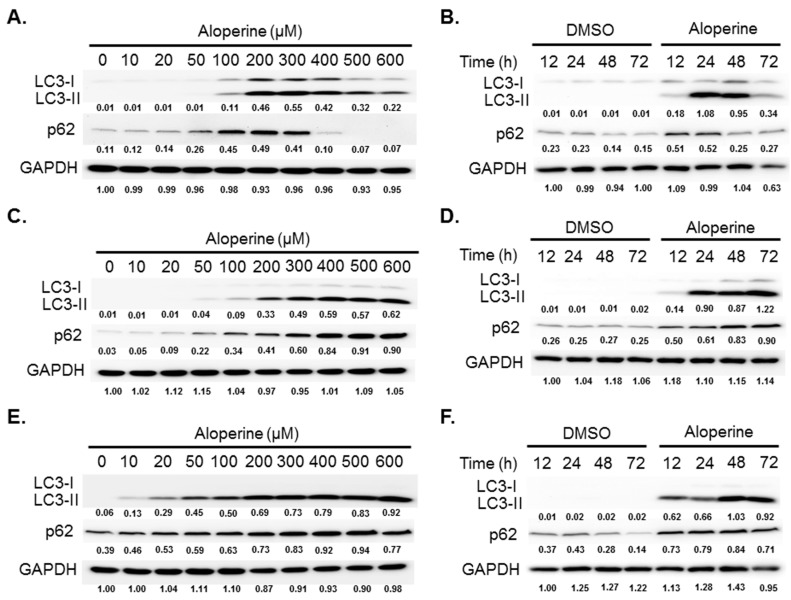
Autophagy modulation in human thyroid cancer cells following aloperine treatment. (**A**,**B**) KMH-2, (**C**,**D**) 8505c, and (**E**,**F**) IHH-4 cells were incubated with aloperine, and LC3-II and p62 expression was determined using western blotting (**A**,**C**,**E**). Data are shown for cells incubated with aloperine for 24 h. (**B**,**D**,**F**) Data are shown for cells treated with DMSO or aloperine (200 µM). Numbers under the plots indicate the quantification of protein intensity after normalization with GAPDH. Three independent experiments were performed, and results of one of these are shown.

**Figure 3 ijms-20-05315-f003:**
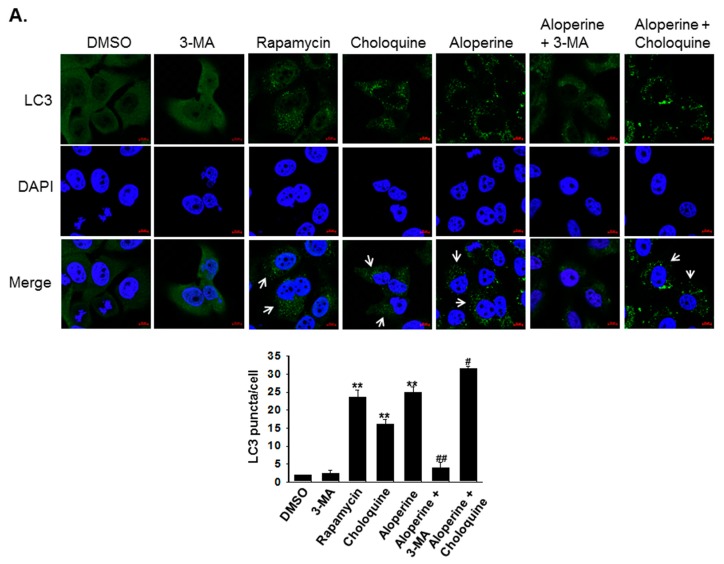
Autophagosome monitoring in human thyroid cancer cells following aloperine treatment. (**A**) KMH-2 and (**B**) IHH-4 cells were incubated with 3-MA (5 mM), rapamycin (30 µM), or aloperine (200 µM) for 24 h, and autophagosome formation was examined using immunofluorescence staining. LC3 (**green**) was used to label autophagosomes and DAPI to label the nucleus (**blue**). DMSO was used as a negative control; 3-MA was used as the autophagy inhibitor and rapamycin as the autophagy inducer. Cells with elevated autophagosome signaling are labeled with an arrow. Autophagosome puncta in the cells were identified and quantified in 30 cells. * *p* < 0.05, ** *p* < 0.01, compared with the DMSO group. ^##^
*p* < 0.01, compared with the aloperine group. Three independent experiments were performed, and results of one of these are shown.

**Figure 4 ijms-20-05315-f004:**
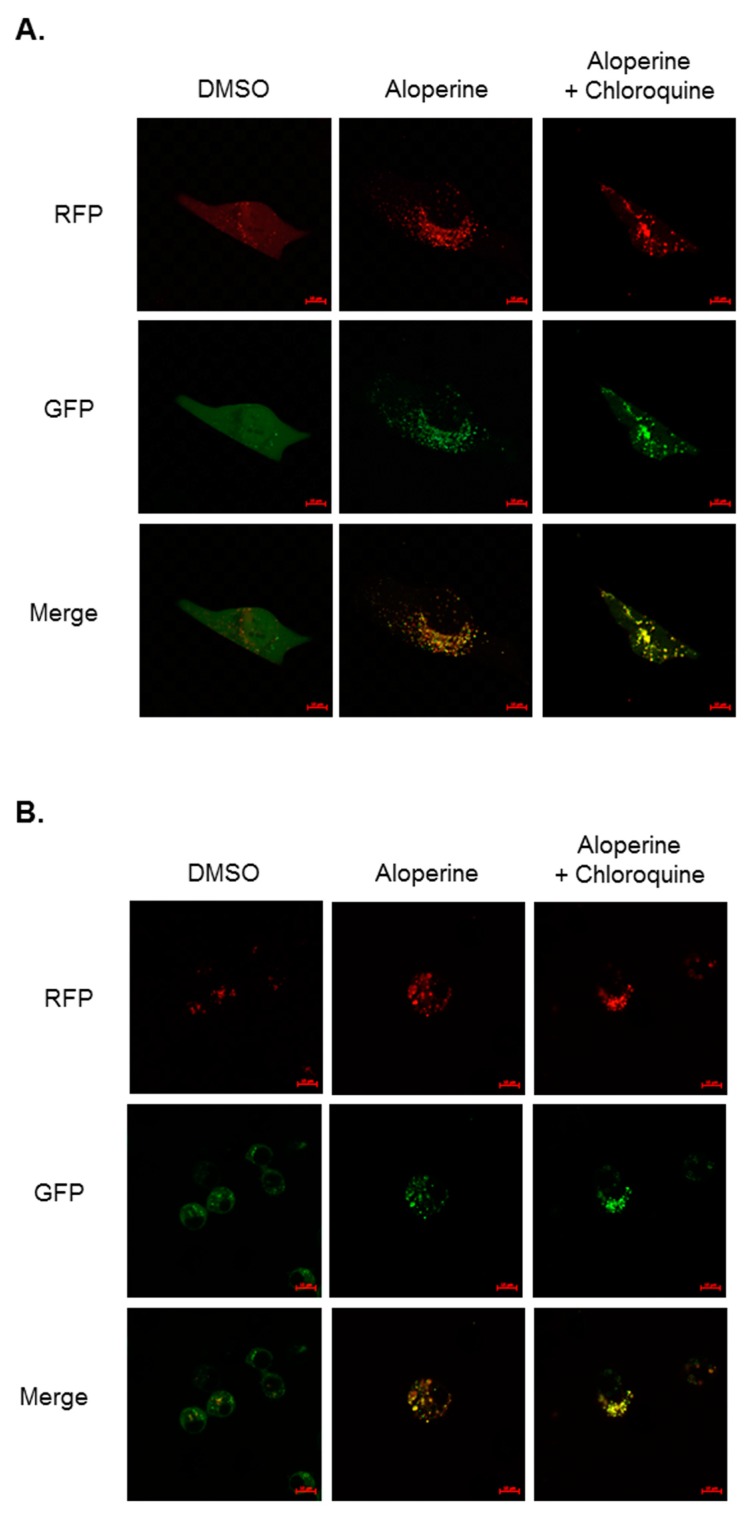
Induction of autophagic flux in human anaplastic thyroid cancer cells following aloperine treatment. (**A**) KMH-2 and (**B**) IHH-4 cells expressing pmRFP-EGFP-LC3 were subjected to aloperine (200 µM) treatment for 24 h with or without 5 µM chloroquine to determine autophagosome/lysosome fusion. Autophagosome (**yellow**) and autolysosome (**red**) puncta increased in the aloperine group, and most puncta were yellow (autophagosome) in the chloroquine combination group. DMSO was used as a negative control.

**Figure 5 ijms-20-05315-f005:**
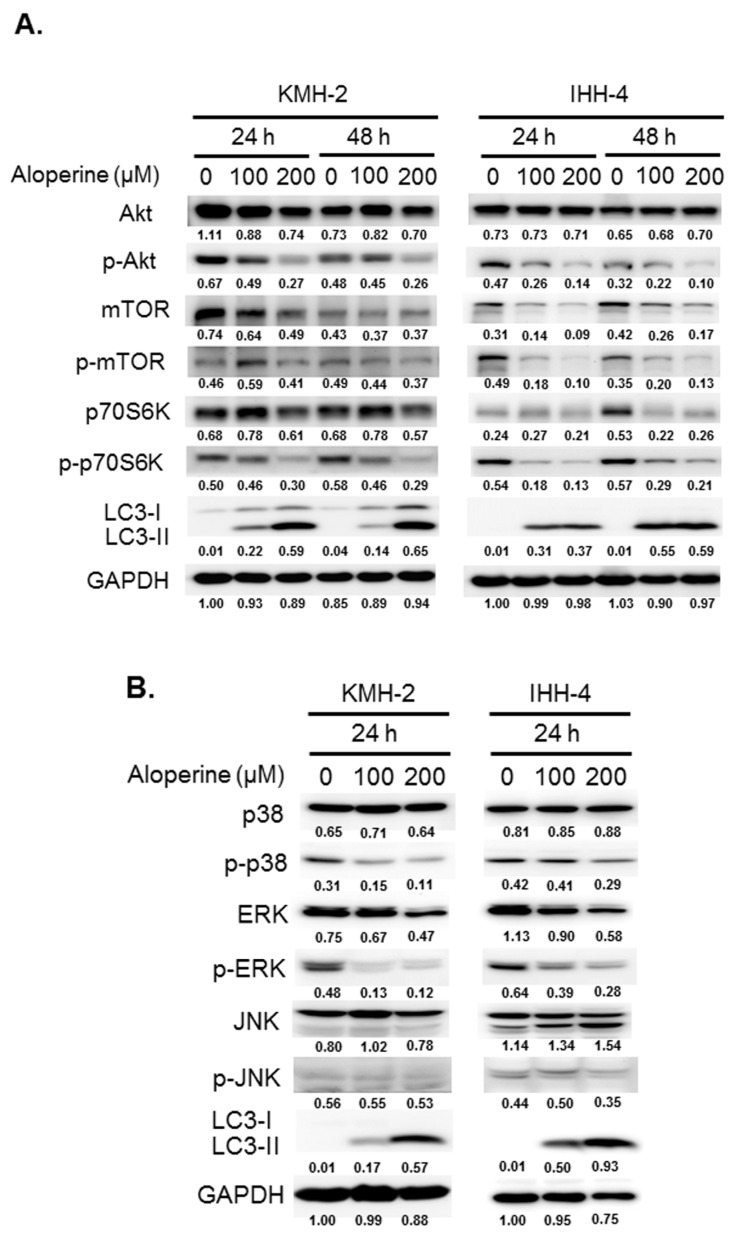
Signaling pathway modulation in human thyroid cancer cells following aloperine treatment. The signaling pathways including (**A**) Akt/mTOR/p70s6K and (**B**) Erk, p38, and JNK pathways were examined in cells with or without aloperine treatment. DMSO was used as a negative control. Numbers under the plots indicate the quantification of protein intensity after normalization with GAPDH. Three independent experiments were performed, and results of one of these are shown.

**Figure 6 ijms-20-05315-f006:**
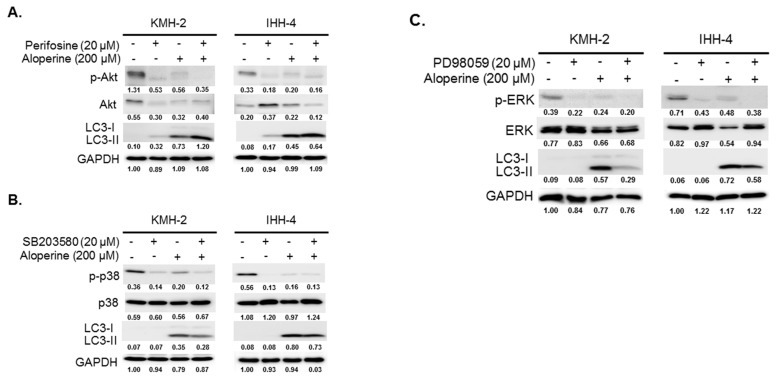
Evaluation of signaling pathways involved in aloperine-mediated autophagy induction in human thyroid cancer cells. To confirm the signaling pathways involved in aloperine-mediated autophagy induction, cells were treated with (**A**) perifosine (an Akt inhibitor), (**B**) SB203580 (a p38 inhibitor), or (**C**) PD98059 (an Erk inhibitor) with or without aloperine; LC3, phospho-Akt, phospho-p38, and phospho-Erk expression was examined using Western blotting after 24 h incubation. DMSO was used as a negative control. Numbers under the plots indicate the quantification of protein intensity after normalization with GAPDH. Three independent experiments were performed, and results of one of these are shown.

**Figure 7 ijms-20-05315-f007:**
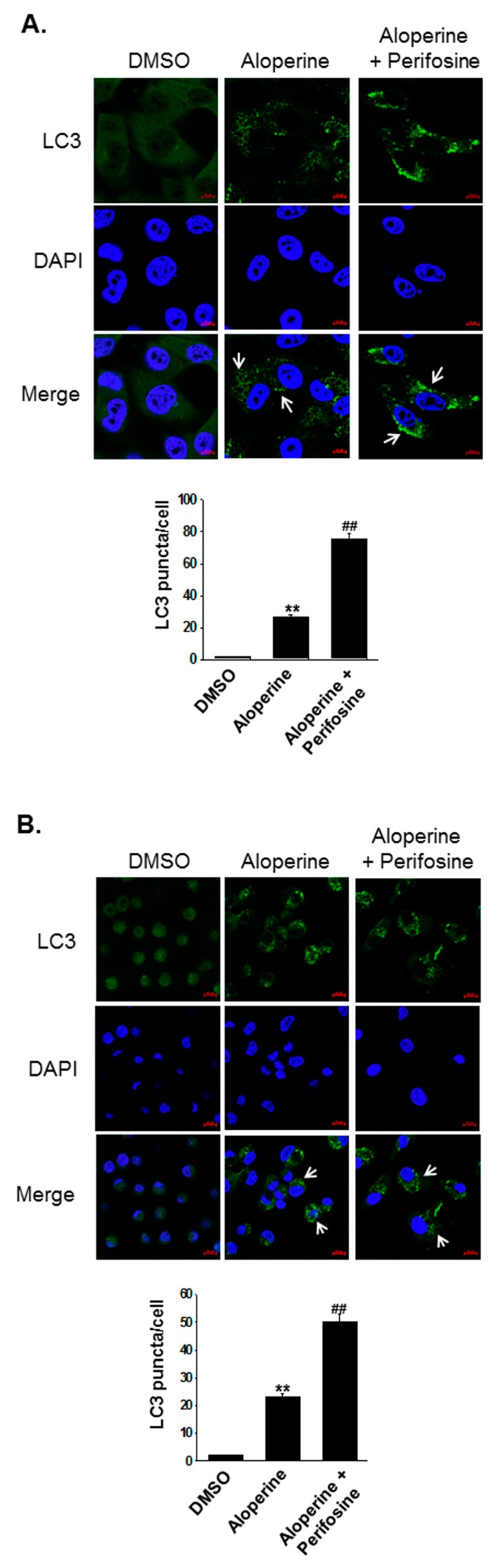
Promotion of aloperine-mediated autophagosome formation via the suppression of Akt pathway activation in human thyroid cancer cells. (**A**) KMH-2 and (**B**) IHH-4 cells were treated with aloperine (200 µM) alone or in combination with perifosine (20 µM) for 24 h, and autophagosome formation was examined with immunofluorescence staining (**green**). DAPI was used to label the nucleus (**blue**). DMSO was used as a negative control. Cells with elevated autophagosome signaling are labeled with an arrow. Autophagosome puncta in cells were identified and quantified in 30 cells. ** *p* < 0.01, compared with the DMSO group. ^##^
*p* < 0.01, compared with the aloperine group. Three independent experiments were performed, and results of one of these are shown.

**Figure 8 ijms-20-05315-f008:**
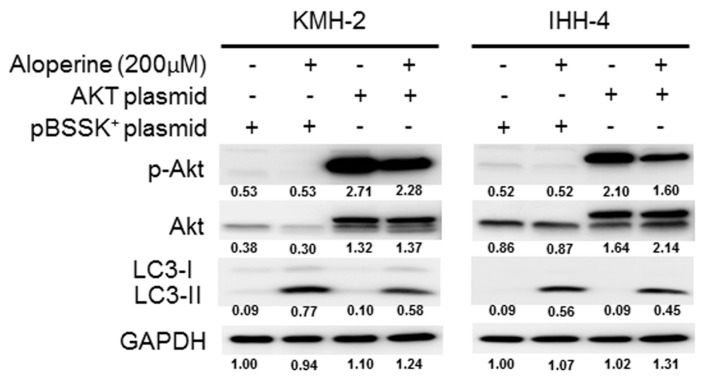
Suppression of aloperine-mediated autophagy induction via compensation of phospho-Akt. Human thyroid cancer cells were transfected with a constitutively active-form Akt construct or a vacant construct. Phospho-Akt and LC3 expression in cells with or without aloperine treatment for 24 h was examined using Western blotting. DMSO was used as a negative control. Numbers under the plots indicate the quantification of protein intensity after normalization with GAPDH. Three independent experiments were performed, and results of one of these are shown.

**Figure 9 ijms-20-05315-f009:**
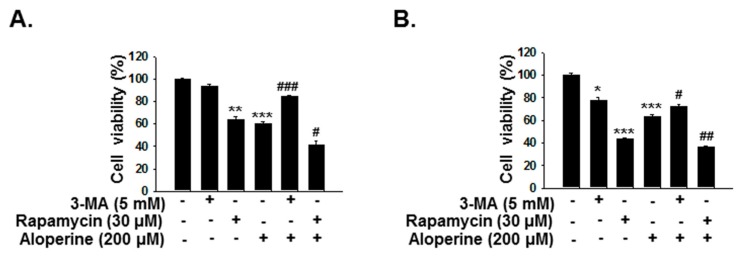
Modulation of autophagy in human thyroid cancer cells affects aloperine-mediated cytotoxicity. (**A**) KMH-2, and (**B**) IHH-4 cells were incubated with aloperine, rapamycin, 3-MA alone, or combination treatment, and their viabilities were determined after 24 h post-treatment using the CCK-8 assay. Cells treated with DMSO were used as a negative control, and all groups were normalized with the control group. The results are expressed as mean ± SD of three independent experiments.

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
