# Peer review of "Autophagy Modulation in Human Thyroid Cancer Cells following Aloperine Treatment"

_ijms, 2019, doi:10.3390/ijms20215315_

Round 1
Reviewer 1 Report
The authors have addressed all my concerns.
Reviewer 2 Report
IJMS_2019
Title: Autophagy Modulation in Human Thyroid Cancer Cells after Aloperine Treatment
This paper investigated the autophagy modulation of Aloperine in human thyroid cancer cells. They previous exhibited that aloperine exerts antitumor effects on human thyroid cancer cells through anti-tumorigenesis and caspase-dependent apoptosis induction via the Akt signaling pathway. In the present study, they reported that incubating multidrug-resistant papillary and anaplastic human thyroid cancer cells with aloperine can cause modulation of the autophagy mechanism; the underlying mechanisms, including AMPK, Erk, JNK, p38 and Akt signaling pathways. This paper clearly revised my comments. However, a few data are not clear to exactly understood the anticancer mechanism of aloperine.
Abatract: Authors should be concise the abstract according to your results. Figure 1, the resolution of x and y-axis very poor. Revised this figure with increased font size. Figure 1, Whati means n=6, “The results are expressed as mean ± SD (n = 6) of three independent experiments” In Figure 2. Modulation of autophagy in human thyroid cancer cells after aloperine administration. Did you did duplicate experiments? Authors must be indicated only one figure. Figure 3,4,5,6,7,8 delete one figure. References. Revised according to the Journal formatting.
Reviewer 3 Report
The authors have performed some of the suggested experiments and amended some of the text. However, in my opinion, still some work is needed to clarify several key findings that lack clear explanations, as well as to improve the quality of some of the figures. Below are comments refer to results interpretation and suggestions to improve the manuscript.
In this new version the authors do not show the IC50 values for normal human fibroblasts (276 microM). I agree with that and therefore, in the new version is not stated that aloperine is selective for cancer cells. On the other hand, the results showed for 805c (268 microM) are similar to those of KHM-2 (207 microM) and IHH4 (169), and therefore the authors should not conclude that “805c is more resistant for the treatment with aloperine”. Again, the quality of the figures 2 and 6 showing the corresponding immunofluorescence stain is very poor, making it impossible to draw conclusions. The authors should enlarge these figures, specially the fields (i.e. showing 3-4 cells instead of 10-20 cells), so the change of LC3 staining into a punctuate pattern is undoubtedly observed. This applies for Figure 2 and 6. Also, all immunofluorescence figures lack the corresponding bar scale. An important point of this work is to show that aloperine induces autophagic flux. This conclusion should be accomplished after performing different types of experiments. The authors show a new figure using immunostaining of mRFP-EGFP-LC3 for a single KHM-2 cell. They should carry out same type of experiment using IHH4 cells. However, this figure does not clearly show the existence of autophagic flux per se. I would recommend the authors to carry out an extra experiment for this purpose. They should show that blocking autophagic flux results in a clear increase of both LC3-II and p62 in by immunoblot analysis. This blocking can be performed, for instance, either using the lysosomal protease inhibitors E64d and Pepstatin-A or the bafilomycin A antibiotic, which prevents maturation of autophagic vacuoles by inhibiting fusion between autophagosomes and lysosomes. As I stated in my first report, the paper will substantially improve if the authors address the question about the role of autophagy in the cytotoxic effect of aloperine in cancer cells. Those experiments are missing in the new version. In this new version the authors included the quantification of the signal of immunoblots, which helps to show that aloperine induces inhibition of the Akt-mTORC1 axis. However, I have concerns s about the study on the other canonical regulator of autophagy AMPK (together with mTORC1). Specifically, in this new version the authors show (not stated before) the use of the antibody that recognizes the phosphorylation of Ser485 to monitor activity of AMPK. The role of this site in AMPK in still controversial. Instead, the authors should use the canonical anti-phospho-Thr172 antibody to check the AMPK activity: to check if aloperine induces of AMPK by promoting phosphorylation of Thr172, which might result in activation of autophagy through ULK phosphorylation). Again, some information about the phoshospecific antibodies used is missing. Authors should state in Methods Sections the specific residues recognized by those antibodies (for instance, which antibody was used to monitor Akt phosphorylation, anti- Ser473 or Thr308?) The manuscript would benefit from a thorough proof-reading for English as there are grammar inaccuracies/errors throughout the text.Author Response
Please see the attachment.

Round 2
Reviewer 3 Report
See comments for editor
Author Response
Please see the attachment.

This manuscript is a resubmission of an earlier submission. The following is a list of the peer review reports and author responses from that submission.
Round 1
Reviewer 1 Report
In this work the authors establish that Aloperine, a natural alkaloid with anticancer activity, induces autophagy in multidrug-resistant human thyroid cancer cells. They also show that Aloperine alters basal levels of active AMPK, Akt/mTORC1, ERK1/2 and stress kinases. On the above basis, the authors conclude that autophagy might mediate the cytotoxic effect of Aloperine in cancer cells.
Identifying anticancer compounds that exert their action through activating cellular autophagy is relevant, since this strategy has been proved to work in vivo, and few pro-autophagy drugs with anticancer activity are now in Clinical Trials. However, this a very descriptive study that lacks mechanistic studies. The authors used a limited number of techniques, and I have concerns about the quality of the figures: I am afraid the experimental data do not support the main conclusions. In this regard, the manuscript will benefit if the authors could increase the quality of some of the figures. Also, Methods section lack critical information.
The following concerns need to be addressed:
The authors very convincingly show that aloperine treatment of cancer cells results in an increase of the lipidated LC3-II form (immunoblot analysis). As no individual assay is enough to properly monitor autophagy, the authors also performed immunofluorescence assay to monitor the staining of LC3. This experiment should clearly show the change from a homogenous LC3 stain in basal conditions, to a punctuate pattern in response to aloperine (lipidated LC3 redistributes to autophagosomes). However, the quality of the figures showing the corresponding immunofluorescence stain is very poor, making it impossible to draw conclusions. The authors should show enlarged figures where a change to punctuate pattern in response to aloperine treatment could be undoubtedly observed. This applies for Figure 2 and 6. How the experiments with cells were performed? Were the cells serum-starved? The authors should clarify whether aloperine activates autophagy per se, or it exacerbates serum-deprivation induced autophagy After performing cell viability assays, the authors conclude that aloperine is highly selected for cancer cells. This is an important conclusion. However, the IC50 values obtained for Hs68 normal human fibroblasts (276 microM) are very similar to those obtained for thyroid cancer cells lines such as 8505c (268 microM). The authors should clarify this point, or provide new data supporting their conclusion. Also, and in order to clarify the role of autophagy in the antitumoral action of aloperine, it would be interesting to study if this compound activates or not autophagy in non-tumoral cells. Also, a very similar Figure 1 was published before in the previous work published by the authors in IJMS (IJMS 19:312, 2018)… The last step of autophagy consists in the fusion of the autophagosome with a lysosome, where autophagic cargos will be degraded and degradation products will be released back to the cytosol. An accumulation of autophagosomes does not necessarily indicate a higher level of autophagy. Therefore, it is necessary to evaluate the whole autophagic flux (dynamic autophagy) to make sure that autophagy is being induced. To do so, the authors simply monitoring the levels of the cargo protein p62 conclude that aloperine induces autophagic flux. This is no correct. The authors should show that blocking fusion with lysosomes or inhibiting lisosomal activity (for example, with lysosomal protease inhibitors E64d and Pepstatin-A) results in enhanced LC3-II accumulation (immunoblot) and puncta (immunofluorescence). The paper will substantially improve if the authors address the question about the role of autophagy in the cytotoxic effect of aloperine in cancer cells. Which is the relationship between the autophagy observed and the aloperine-induced cytotoxicity? Does aloperine induce an autophagy-mediated apoptosis?. To test this, the authors should investigate which is the effect of pharmacological inhibition (E64d + Pepstatin A) or of genetical inhibition (ATG5 silencing) on the cytotoxic (MTT assay) effect of aloperine in cancer cells. Using immunoblot analysis, the authors conclude aloperine treatment results in inhibition of AMPK and Akt/mTOR/p70S6K pathways. Again, I have concerns about the quality of this figure (Figure 4). Fist, the authors should state in Methods Sections information about the phosphospecific antibodies used (that info is missing). Total levels of mTOR protein diminishes with aloperine treatment, in the same fashion as phospho-mTOR signal does. This also applies for pS6K/S6K blots for the IHH-4 cells. This blot should be repeated to show equal levels of total proteins, so the reader can notice changes in the phosphorylated state of these proteins. The authors should also comment that aloperine treatment induces reduced expression levels of the mTOR and S6K proteins. Also, Figure 4 upper lane shows total levels of regulatory subunit AMPK beta-1, whereas the second lane shows the levels of phosphorylated (Thr172?) catalytic subunit AMPK alfa. This is not correct. In order to draw conclusions about the activation state of AMPK, the authors must show total levels of AMPK alfa subunit. I have serious concerns about the described role of Akt in the mechanism of action of aloperine. Figure 5 shows that inhibition of Akt with 20 microM perifosine results in a very weak activation of autophagy (LC3-II), much lower than that showed for aloperine. This result, which might show that other pathways than Akt could be involved in the mechanism of action of aloperine, should be mentioned and discussed in the text. Furthermore, Figure 5 shows changes on Akt protein levels in response to 24 h aloperine treatment, whereas Figure 4 shows no changes on Akt levels upon same treatment. The authors should clarify this. Figure 7 shows the results on levels of autophagy in response to overexpressing a constitutively active form of Akt. First, the authors should state in the Methods section that this is the myristoylated form of Akt-1 (that information is missing). The authors concluded that “over-expression of active form-Akt in cells treated with aloperine reduce the expression of LC3-II, demonstrating that Akt signaling pathway is the upstream pathway in the aloperine-mediated autophagy induction”. However, Figure 7 does not show this, but rather shows no changes on LC3-II levels. The authors should clarify this important point, since both experiments (pharmacological inhibition and active form overexpression) point to the fact that Akt plays a minor role in the autophagy induced by aloperine in cancer cells. The manuscript would benefit from a thorough proof-reading for English as there are numerous inaccuracies/errors throughout the text. The expression “constantly active” should be changed to “constitutively active”
Author Response
Explanations to the queries of Reviewer 1
Query No 1: The authors very convincingly show that aloperine treatment of cancer cells results in an increase of the lipidated LC3-II form (immunoblot analysis). As no individual assay is enough to properly monitor autophagy, the authors also performed immunofluorescence assay to monitor the staining of LC3. This experiment should clearly show the change from a homogenous LC3 stain in basal conditions, to a punctuate pattern in response to aloperine (lipidated LC3 redistributes to autophagosomes). This applies for Figure 2 and 6.
Reply to query 1: The quantification of autophagy puncta in the cells has been done and illustrated in the figures. Moreover, the western blot data were also quantized and explained in the figures.
Query No 2: How the experiments with cells were performed? Were the cells serum-starved?
Reply to query 2: The cells in all experiments are without starvation.
Query No 3: After performing cell viability assays, the authors conclude that aloperine is highly selected for cancer cells. This is an important conclusion. However, the IC50 values obtained for Hs68 normal human fibroblasts (276 microM) are very similar to those obtained for thyroid cancer cells lines such as 8505c (268 microM). The authors should clarify this point, or provide new data supporting their conclusion.
Reply to query 3: It suggests that KMH-2 and IHH-4 cells are more sensitive for aloperine treatment; however, 8505c cells are more resistance for the treatment with aloperine. Whether the difference in 8505c cells resistance is due to genetic background or other reasons still needs further investigation.
Query No 4: In order to clarify the role of autophagy in the antitumoral action of aloperine, it would be interesting to study if this compound activates or not autophagy in non-tumoral cells.
Reply to query 4: Autophagy induction in these situations exercises either a cytotoxic or cytoprotective role (Cancer Res 74, no. 3 (2014): 647-51), demonstrating that the induction of autophagy in cancer cells may be more complicated. Although excessive and sustained autophagy may lead to cell death and tumor shrinkage, and autophagic cell death has been reported in numerous reports, the cytotoxic role of autophagy remains under discussion because of the insufficient data on autophagic cell death markers (Biomed Res Int 2015 (2015): 934207).
Query No 5: The last step of autophagy consists in the fusion of the autophagosome with a lysosome, where autophagic cargos will be degraded and degradation products will be released back to the cytosol. An accumulation of autophagosomes does not necessarily indicate a higher level of autophagy. Therefore, it is necessary to evaluate the whole autophagic flux (dynamic autophagy) to make sure that autophagy is being induced.
Reply to query 5: To confirm the autophagic flux induction in cancer cells under aloperine treatment, a pmRFP-EGFP-LC3 construction was transfected into KMH-2 cells and the autophagosome/autolysosome were determined in Figure 4.
Query No 6: Which is the relationship between the autophagy observed and the aloperine-induced cytotoxicity? Does aloperine induce an autophagy-mediated apoptosis?
Reply to query 6: Autophagy induction in these situations exercises either a cytotoxic or cytoprotective role (Cancer Res 74, no. 3 (2014): 647-51), demonstrating that the induction of autophagy in cancer cells may be more complicated. Although excessive and sustained autophagy may lead to cell death and tumor shrinkage, and autophagic cell death has been reported in numerous reports, the cytotoxic role of autophagy remains under discussion because of the insufficient data on autophagic cell death markers (Biomed Res Int 2015 (2015): 934207). In the present study, we are the first to demonstrate that aloperine can modulate the autophagy machinery and induce autophagosome as well as autophagic flux in human thyroid cancer cells. The cytotoxic or cytoprotective role of aloperine-mediated autophagy needs further evaluation.
Query No 7: Authors should state in Methods Sections information about the phosphospecific antibodies used (that info is missing).
Reply to query 7: The information of antibodies has been added in the methods section (anti-LC3 Ab; Abcam, USA; anti-GAPDH Ab; GeneTex, USA; anti-AMPK α-1 Ab; Cell signaling, USA; anti-phosphorylated-AMPK α-1 Ab; Cell Signaling, USA; anti-Akt Ab; Santa Cruz, USA; anti-phosphorylated-Akt Ab; Santa Cruz, USA; anti-mTOR Ab; Cell Signaling, USA; anti-phosphorylated-mTOR Ab; Cell Signaling, USA; anti-p70S6K Ab; Cell Signaling, USA; anti-phosphorylated-p70S6K Ab; Cell Signaling; anti-p62/SQSTM1 Ab; Abgent, USA; anti-Erk Ab; Cell Signaling, USA; anti-phosphorylated-Erk Ab; Cell Signaling, USA; anti-JNK Ab; Cell Signaling, USA; anti-phosphorylated-JNK Ab; Cell Signaling, USA; anti-p38 Ab; Cell Signaling, USA; and anti-phosphorylated Ab; Cell Signaling Ab, USA.).
Query No 8: The authors should also comment that aloperine treatment induces reduced expression levels of the mTOR and S6K proteins.
Reply to query 8: The western blot data has been quantized and explained in the figures. In addition, under the treatment of aloperine, the expression of total mTOR showed a decrease (Figure 5A), indicating that aloperine may regulate the translation of mTOR.
Query No 9: Figure 4 upper lane shows total levels of regulatory subunit AMPK beta-1, whereas the second lane shows the levels of phosphorylated (Thr172?) catalytic subunit AMPK alfa. This is not correct. In order to draw conclusions about the activation state of AMPK, the authors must show total levels of AMPK alfa subunit.
Reply to query 9: The phospho-AMPK-α1 (Ser 485) Ab and the Western blot data of total AMPK-α1 have been revised.
Query No 10: I have serious concerns about the described role of Akt in the mechanism of action of aloperine. Figure 5 shows that inhibition of Akt with 20 microM perifosine results in a very weak activation of autophagy (LC3-II), much lower than that showed for aloperine. This result, which might show that other pathways than Akt could be involved in the mechanism of action of aloperine.
Reply to query 10: The explanation “In addition, we cannot rule out whether factors excluding Akt pathway are involved in aloperine mediated autophagy.” has been included in the text.
Query No 11: Figure 7 shows the results on levels of autophagy in response to overexpressing a constitutively active form of Akt. First, the authors should state in the Methods section that this is the myristoylated form of Akt-1 (that information is missing).
Reply to query 11: The explanation “To confirm the role of Akt pathway in aloperine-mediated autophagosome induction, plasmid pHRIG-Akt1 (a constitutively active-form construction of human Akt1, a myristoylated form of Akt-1, purchased from Addgene) and pBSSK+ (an empty vector used as a negative control) were transfected with Lipofectamine 2000 (Thermo Scientific, Waltham, MA, USA) according to the manufacturer’s instructions.” has been edited in the methods section.
Query No 12: The expression “constantly active” should be changed to “constitutively active”.
Reply to query 12: It has been edited in the text.
Reviewer 2 Report
ijms-568101
Title: Autophagy Modulation in Human Thyroid Cancer Cells after Aloperine Treatment
This study investigated the anticancer effects of aloperine on authohapy cell death in thyroid cancer cells. Based on these results, authors demonstrated that aloperine exerts antitumor effects on human thyroid cancer cells through anti-tumorigenesis and caspase-dependent apoptosis induction via the Akt signaling pathway. This results are well described but a few data are not clear to understanding the molecular mechanism of aloperine on the thyroid cancer death pathways.
Major
Xu et al (2017) previously indicated that reducing autophagy and inducing G1 phase arrest by aloperine enhances radio-sensitivity in lung cancer cells. In addition, your previous data already showed that aloperine induces caspase-dependent apoptosis through the inhibition of PI3K/Akt pathway in human thyroid cancer cells [15]. In the present study, cells treated with aloperine demonstrated the induction of autophagy through suppression of Akt/mTOR pathway in human thyroid cancer cells. Which cell death pathway is more pronounced by aloperine treatment in thyroid cancer cells. If you both pathways are involved the aloperine-mediated anticancer effects, authors must be compared the cell death pathways using Akt inhibitors in thyroid cancer cells. In Introduction part, aloperine exhibits anti-inflammatory, antibacterial, antiviral, and antitumor activity. In addition, increases p21 and p53 and decreases cyclin D1 and B1 levels. Induces G2/M phase cell cycle arrest and apoptosis. As you indicated that autophagy cell death is closely related with AMPK activation, thus aloperine directly effects on the AMPK activation? As you showed in Figure 4A, the activation of the AMPK pathway was reduced in IHH-4 cells, but no change was observed in KMH-2 cells. Moreover, the activation of Akt/mTOR and p70S6K pathways decreased in a dosage related manner with aloperine treatment in both the cells, whereas the expression of LC3-II increased. For this results, authors must be confirming these results using apoptosis inhibitor in two different type of thyroid cancer cells. As your described in Discussion part, “However, we cannot exclude the possible involvement of caspase-independent apoptosis or autophagy in the aloperine-mediated effects on human thyroid cancers” What is the major anticancer effects of aloperine on thyroid cancer cells.? Authors may confirm again the antitumor activity of aloperine using in vivo tumor xenograft model.
Author Response
Explanations to the queries of Reviewer 2:
Query No 1: Xu et al (2017) previously indicated that reducing autophagy and inducing G1 phase arrest by aloperine enhances radio-sensitivity in lung cancer cells.
Reply to query 1: This article has been withdrawn. Therefore, our study is the first investigation for autophagy induction under aloperine treatment.
Query No 2: your previous data already showed that aloperine induces caspase-dependent apoptosis through the inhibition of PI3K/Akt pathway in human thyroid cancer cells [15]. In the present study, cells treated with aloperine demonstrated the induction of autophagy through suppression of Akt/mTOR pathway in human thyroid cancer cells. Which cell death pathway is more pronounced by aloperine treatment in thyroid cancer cells. If you both pathways are involved the aloperine-mediated anticancer effects, authors must be compared the cell death pathways using Akt inhibitors in thyroid cancer cells. What is the major anticancer effects of aloperine on thyroid cancer cells? Authors may confirm again the antitumor activity of aloperine using in vivo tumor xenograft model.
Reply to query 2: Autophagy induction in these situations exercises either a cytotoxic or cytoprotective role (Cancer Res 74, no. 3 (2014): 647-51), demonstrating that the induction of autophagy in cancer cells may be more complicated. Although excessive and sustained autophagy may lead to cell death and tumor shrinkage, and autophagic cell death has been reported in numerous reports, the cytotoxic role of autophagy remains under discussion because of the insufficient data on autophagic cell death markers (Biomed Res Int 2015 (2015): 934207). In the present study, we are the first to demonstrate that aloperine can modulate the autophagy machinery and induce autophagosome as well as autophagic flux in human thyroid cancer cells. The cytotoxic or cytoprotective role of aloperine-mediated autophagy needs further evaluation. In addition, aloperine exhibits multiple pharmacological activities, including antiinflammatory, antiallergenic, antiviral, antimicrobial, antinociceptive, against renal and neuronal injury and pulmonary fibrosis. Here, we demonstrate that aloperine can regulate autophagy; it is interesting to further investigate whether aloperine can protect renal and neuronal injury or pulmonary fibrosis through autophagy induction.
Reviewer 3 Report
In the manuscript entitled “Autophagy Modulation in Human Thyroid Cancer Cells after Aloperine Treatment” the authors have reported that incubating multidrug resistant papillary and anaplastic human thyroid cancer cells with aloperine can cause modulation of the autophagy mechanism; the underlying mechanisms, including AMPK, Erk, JNK, p38 and Akt signaling pathways, are also illustrated. Further investigation demonstrated that Akt signaling pathway is involved in the aloperine-modulated autophagy in human thyroid cancer cells. These results demonstrate a previously unappreciated function of aloperine in autophagy modulation in human thyroid cancer cells.
Specific Comments:
This is technically well performed study but the authors need to address several missing links before it can be even considered for publication. Specific points that the authors need to address are as follows:
The molecular mechanism(s) by which aloperine can induce autophagy not clear? For example, whether deletion of Erk, AMPK, p38 and JNK by siRNA can abrogate the observed autophagic effects of aloperine should be analyzed? The effect of aloperine should also be analyzed on normal epithelial cells to rule out potential cytotoxicity. Also, acute toxicity studies should be performed to establish the safety of the compound. A limited in vivo study in appropriate xenograft/orthotopic mouse model is required and will greatly increase the impact of the reported in vitro findings. Western blot data should be quantitated along with statistical analysis. The manuscript should be carefully edited for English. The authors should discuss their own justification and relevance of the study. This will help the readers to understand the importance of the paper otherwise it appears to be just an extension of their previous published work with the same compound.Author Response
Query No 1: The molecular mechanism(s) by which aloperine can induce autophagy not clear? For example, whether deletion of Erk, AMPK, p38 and JNK by siRNA can abrogate the observed autophagic effects of aloperine should be analyzed? The effect of aloperine should also be analyzed on normal epithelial cells to rule out potential cytotoxicity. Also, acute toxicity studies should be performed to establish the safety of the compound. A limited in vivo study in appropriate xenograft/orthotopic mouse model is required and will greatly increase the impact of the reported in vitro findings. Western blot data should be quantitated along with statistical analysis. The manuscript should be carefully edited for English. The authors should discuss their own justification and relevance of the study. This will help the readers to understand the importance of the paper otherwise it appears to be just an extension of their previous published work with the same compound.
Reply to query 1: Thanks for the comments. Although our results show that aloperine can reduce the activation of the p38 and Erk pathways in both KMH-2 and IHH-4 cells (Figure 5A and 5B), enhancing the suppression of the p38 and Erk activations with SB203580 and PD98059 in the cells under aloperine treatment does not significantly increase LC3-II levels (Figure 6B and 6C). These results suggest that aloperine-mediated inhibition of p38 and Erk pathways is not the underlying mechanism for aloperine-mediated autophagy induction. However, the physiological significance of aloperine-inhibited p38 and Erk signaling pathways in KMH-2 and IHH-4 cells needs further investigation. In addition, we cannot rule out whether factors excluding Akt pathway are involved in aloperine mediated autophagy. In addition, an in vivo study has demonstrated the safety and efficacy of aloperine as a therapeutic agent (J Hematol Oncol 8 (2015): 26.); therefore, the issue of toxicity of aloperine in normal cells or in vivo is suggesting to be safe. In the present study, we are the first to demonstrate that aloperine can modulate the autophagy machinery and induce autophagosome as well as autophagic flux in human thyroid cancer cells. In addition, aloperine exhibits multiple pharmacological activities, including antiinflammatory, antiallergenic, antiviral, antimicrobial, antinociceptive, against renal and neuronal injury and pulmonary fibrosis. Here, we demonstrate that aloperine can regulate autophagy; it is interesting to further investigate whether aloperine can protect renal and neuronal injury or pulmonary fibrosis through autophagy induction.